# Effects of Hibernation on Colonic Epithelial Tissue and Gut Microbiota in Wild Chipmunks (*Tamias sibiricus*)

**DOI:** 10.3390/ani14101498

**Published:** 2024-05-17

**Authors:** Juntao Liu, Guangyu Jiang, Hongrui Zhang, Haiying Zhang, Xiaoyan Jia, Zhenwei Gan, Huimei Yu

**Affiliations:** 1College of Basic Medical Sciences, Jilin University, Changchun 130021, China; jtliu2720@mails.jlu.edu.cn (J.L.); jianggy2720@mails.jlu.edu.cn (G.J.); zhanghr8220@mails.jlu.edu.cn (H.Z.); zhanghaiy@jlu.edu.cn (H.Z.); jiaxiaoyan@jlu.edu.cn (X.J.); 2School of Public Health, Jilin University, Changchun 130021, China; ganzw@jlu.edu.cn

**Keywords:** wild chipmunk (*Tamias sibiricus*), gut microbiota, hibernation, goblet cell, high-throughput sequencing

## Abstract

**Simple Summary:**

Wild chipmunks (*Tamias sibiricus*) eat and drink intermittently during hibernation. Instead of simply storing fat or food before hibernation, they use a mixed hibernation strategy. This experiment studied the histology of the colon, as well as changes in the composition and function of the gut microbiota of wild chipmunks during induced hibernation, aiming to explore how the gut microbiota of wild chipmunks performs its functions to better maintain normal physiological functions of the body during induced hibernation. The findings indicated that hibernation caused the increase of goblet cells in the colonic epithelial tissue of wild chipmunks, and increased the richness and diversity of the gut microbiota. The composition and function of the gut microbiota changed a lot, which could regulate the physiological metabolism of wild chipmunks during hibernation and better maintain their normal physiological functions.

**Abstract:**

The gut microbiota plays a crucial role in the host’s metabolic processes. Many studies have shown significant changes in the gut microbiota of mammals during hibernation to adapt to the changes in the external environment, but there is limited research on the colonic epithelial tissue and gut microbiota of the wild chipmunks during hibernation. This study analyzed the diversity, composition, and function of the gut microbiota of the wild chipmunk during hibernation using 16S rRNA gene high-throughput sequencing technology, and further conducted histological analysis of the colon. Histological analysis of the colon showed an increase in goblet cells in the hibernation group, which was an adaptive change to long-term fasting during hibernation. The dominant gut microbial phyla were Bacteroidetes, Firmicutes, and Proteobacteria, and the relative abundance of them changed significantly. The analysis of gut microbiota structural differences indicated that the relative abundance of *Helicobacter typhlonius* and *Mucispirillum schaedleri* increased significantly, while *unclassified Prevotella-9*, *unclassified Prevotellaceae-UCG-001*, *unclassified Prevotellaceae-UCG-003* and other species of *Prevotella* decreased significantly at the species level. Alpha diversity analysis showed that hibernation increased the diversity and richness of the gut microbiota. Beta diversity analysis revealed significant differences in gut microbiota diversity between the hibernation group and the control group. PICRUSt2 functional prediction analysis of the gut microbiota showed that 15 pathways, such as lipid metabolism, xenobiotics biodegradation and metabolism, amino acid metabolism, environmental adaptation, and neurodegenerative diseases, were significantly enriched in the hibernation group, while 12 pathways, including carbohydrate metabolism, replication and repair, translation, and transcription, were significantly enriched in the control group. It can be seen that during hibernation, the gut microbiota of the wild chipmunk changes towards taxa that are beneficial for reducing carbohydrate consumption, increasing fat consumption, and adapting more strongly to environmental changes in order to better provide energy for the body and ensure normal life activities during hibernation.

## 1. Introduction

The gut microbiota is a dynamic, abundant, and highly diverse microbial community residing in the gastrointestinal tract, including various anaerobic bacteria, fungi, and other microorganisms [1]. The gut microbiota is deemed to be a major factor influencing health [2], as its metabolic products significantly impact the host’s metabolism [3], immune function [4], nervous system [5], energy balance [6], and other aspects related to body health. The gut microbiota forms a symbiotic system with its host, which is established at birth [7], and the host provides a nutrient-rich environment for the gut microbiota that supports the development of diverse microbial communities dwelling in multiple ecological niches [8]. Studies have shown that dietary intervention [9], environmental factors [10], pathogenic bacteria infection [11], circadian rhythm change [12], and the use of dietary supplements [13] can all affect the composition, structure, and function of the gut microbiota. In recent years, the field of research has increasingly involved whether the hibernation rhythm of certain mammals affects their gut microbiota. Mammalian hibernation refers to the physiological state in which animals enter a dormant state to deal with environmental changes, interspersed with brief periods of arousal [14]. During hibernation, the body temperature of animals can drop below 5 degrees, the heart rate and respiratory rate decrease significantly, and the metabolic rate can be reduced by more than 90%, which is conducive to saving energy during the winter when food is scarce [15,16]. The phenomenon of food scarcity caused by hibernation can influence the gut microbiota of animals, and some microbial communities change significantly before and after hibernation. These changes in gut microbiota can in turn affect the immune system, metabolic processes, and intestinal barrier function of the host [17]. Previous studies have found that hibernation could affect the gut microbiota of the 13-lined ground squirrels, altering the abundance, composition, and diversity of gut microbiota and influencing its metabolic products [18]. Hibernation alters the composition and function of gut microbiota in the Daurian ground squirrel [19,20], and affects the barrier function of the colon [21]. Hibernation affects the diversity and composition of the cecal microbiota in Arctic ground squirrels [22], and pre-hibernation diet may regulate the function of the gut microbiota in Arctic ground squirrels [23]. Hibernation affects the composition of the gut microbiota in Syrian hamsters [24]. Similarly, the effects of hibernation on the gut microbiota have been found in other mammals, such as dwarf lemurs [25], brown bears [26], and bats [27]. Among amphibians, for example, studies of the Japanese wrinkled frogs (*Glandirana rugosa*) [28] *Strauchbufo raddei* [29] and *Fejervarya limnocharis* [30] have also revealed the impact of hibernation on the gut microbiota.

The chipmunk, a ground-dwelling squirrel, comprises 25 different species classified into three different genera: the *Eutamias*, the *Tamias*, and the *Neotamias* [31]. The wild chipmunk subject of this study belongs to the *Tamias sibiricus* distributed in northeast China, which is native to northeast Asia [32]. It has established exotic populations in other countries through the pet trade and is naturally distributed in most parts of Eurasia and North America, such as France, Switzerland, Russia, China, Mongolia, and other countries [33,34]. Wild chipmunks eat and drink intermittently during hibernation [35]; instead of simply storing fat or food before hibernation, they use a mixed hibernation strategy [36], which is different from the hibernation pattern of other mammals, most of which do not eat and drink during hibernation. Diet is the main factor affecting the gut microbiota [37], so we speculated that this unique hibernation pattern of wild chipmunks might lead to extraordinary changes in the gut microbiota and colonic epithelial tissue. However, few studies have been done on the effects of hibernation on the gut microbiota and colonic epithelial tissue of wild chipmunks. Different from previous studies, this study took wild chipmunks in northeast China as the research subject, and compared the diversity and composition of the colonic microbiota of wild chipmunks in hibernation and non-hibernation stages through 16S rRNA gene high-throughput sequencing technology. The functional changes of the colonic microbiota during distinct periods were predicted and colonic histological analysis was performed. This study improves the understanding of changes in intestinal histology and the composition and function of the gut microbiota during hibernation in wild chipmunks, providing new information for the current research on how mammalian hibernation affects colonic epithelial tissue and gut microbiota.

## 2. Materials and Methods

### 2.1. Animals and Study Design

The wild chipmunks used in this study were approved by the Experimental Animal Ethics Committee of the School of Basic Medical Sciences, Jilin University. In September 2022, six wild chipmunks were captured in Changchun City, Jilin Province of China, and brought back to the Jilin University Animal Biosafety Laboratory on Level 2 for feeding, where they were fed with corn and had free access to food and water. After an adaptation period of 14 days, the wild chipmunks were randomly divided into two groups, the hibernation group (Hib) and the control group (Con), with 3 chipmunks in each group. The control group was kept under room temperature conditions (23 ± 2) °C, while the hibernation group was kept in a constant temperature box at (4 ± 1) °C without light, and induced for 120 days. Pure carbon dioxide was filled into the box for 2 min, and the chipmunks were observed from the observation window for another 2 min, to confirm they had died. After the carbon dioxide was drained, the chipmunks were removed by turning on the euthanasia device. After euthanizing the experimental animals by carbon dioxide inhalation method, the colon tissues of the two groups of wild chipmunks were collected on an ultra-clean table. The excised colon tissues were placed in cryopreservation tubes filled with 4% paraformaldehyde and stored at room temperature. Colonic contents were immediately stored in liquid nitrogen (−80 °C).

### 2.2. Histological Analysis of the Colonic Epithelial Tissue

The paraformaldehyde fixative was washed off the colon tissues, followed by dehydration with various concentrations of alcohol. After thorough dehydration, the colon tissues were placed in solvent mixed with an equal volume of xylene and absolute alcohol for 1 h, then removed and placed in pure xylene for 20 min. After that, they were put into a mold, melted paraffin was poured, and they were cooled and solidified before being removed. The embedded tissues wax blocks were placed on a paraffin microtome and cut into 5 μm thick sections. Then, they were placed in warm water at 40 °C to flatten the sections, affixed to slides, and dried in a 45 °C incubator. Slices were deparaffinized and stained with hematoxylin-eosin (HE) and then they were observed through the microscope (Olympus BX53, Shinjuku, Tokyo, Japan). Slices were deparaffinized and stained with Alcian Blue-Periodic Acid Schiff (AB-PAS) and then observed through the microscope (OLYMPUS, BX53, Japan).

### 2.3. High-Throughput Sequencing of 16S rRNA Gene

A DNA extraction kit (DP304, Tiangen Biotech Co., Ltd., Beijing, China.) was used to extract total DNA from each intestinal sample. The purity and concentration of DNA were detected by microplate reader, and after checking the integrity of the DNA with 1% agarose gel electrophoresis, qualified samples were sent to Beijing Biomarker Biotechnology Co., Ltd., Beijing, China, for sequencing.

After extraction of total DNA, primers were designed according to the conserved region. The primers used were Forward: 5-ACTCCTACGGGAGGCAGCA-3. Reverse: 5-GGACTACHVGGGTWTCTAAT-3. The PCR products were purified, quantified, and homogenized to form a sequencing library. The constructed libraries were first subjected to library quality inspection, and the qualified libraries were sequenced by Illumina NovaSeq 6000, San Diego, CA, USA.

### 2.4. Bioinformatic and Statistical Analysis

First, the original data were quality filtered using Trimmomatic V0.33 [38], then the primer sequence was identified and removed using Cutadapt V1.9.1 [39]. Subsequently, USEARCH V10.0 [40] was used to splice the double-end reads and remove the chimeras (UCHIME V8.1 [41]), and high-quality sequences were finally obtained. USEARCH V10.0 [40] was used to cluster high-quality sequences at the level of 97% similarity, and by default 0.005% of the number of all sequences sequenced was used as a threshold to filter OTUs. The DADA2 [42] method in QIIME2 V2020.6 [43] was used to denoise the data after quality control. Additionally, 0.005% of the number of all sequences sequenced was used as the threshold to filter ASVs, and the naive Bayes classifier was used to annotate the characteristic sequences with Silva as the reference database [44]. According to the standardized OUT, samples were analyzed for alpha and beta diversity using QIIME2 V2020.6 [43] software. The alpha diversity indices included Chao1, ACE, Shannon, Simpson, PD-whole-tree, and Coverage indices, and the significance of differences in alpha diversity indices was verified by Student *t* test. Principal coordinate analysis (PCoA) of Bray–Curtis’s distance matrices was examined by PERMANOVA analysis. A heatmap was drawn up based on the unweighted UniFrac algorithm to obtain the distance matrix between samples, using R language. Species information was annotated taxonomically from phylum level to species level. The Wilcoxon rank sum test was used to analyze the significance of differences in the dominant flora at the phylum and genus levels. Linear discriminant analysis effect size (LEfSe) was used to determine species differences at different taxonomic levels [45].

Based on high-throughput sequences, the microbial function was predicted using PICRUSt2 [46] to obtain the relative abundance of genes in KEGG pathways. The G-TEST and Fisher test methods in STAMP [47] were used to test the significant difference between samples for functional abundance among different samples, and the pairwise *t*-test was used to analyze the functional pathways with significant differences between different groups (*p* < 0.05).

## 3. Results

### 3.1. Effect of Hibernation on Colonic Epithelial Tissue of Wild Chipmunks

The histological results of the colonic epithelial tissue of the wild chipmunks are shown in Figure 1 and Figure 2. Compared with the control group, the number of goblet cells was increased in the colonic epithelial tissue of the wild chipmunks in the hibernation group.

### 3.2. OTU and Alpha Diversity Analysis

A total of 479,614 PE reads were obtained by 16S rRNA gene sequencing of the colonic contents of six wild chipmunks. A total of 444,761 clean reads were obtained after quality control and assemble, and an average of 74,127 clean reads were generated per sample. High-quality sequences were denoised and OTUs were divided. A Venn diagram showed that a total of 5003 OTUs were obtained from the two groups; 3235 unique OTUs were found in the hibernation group and 1751 unique OTUs were found in the control group. The number of OTUs in the hibernation group was more than that in the control group, and 17 OTUs were common between the two groups, accounting for only 0.34% of the total OTUs (Figure 3A). Both rarefaction curves (Figure 3B) and Shannon Index curves (Figure 3C) showed that the curves of all samples were flattening, indicating that the number of sequences was large enough and that the feature species would not grow with the increase of sequencing volume. The sequences could be used for data analysis.

Alpha diversity analysis showed that the diversity and richness of the gut microbiota in the hibernation group were significantly higher than those in the control group. The ACE index and Chao1 index of the hibernation group were significantly higher than those of the control group (*p* < 0.01), indicating that the species richness of the hibernation group was higher (Figure 4A,B). The Shannon index was used to measure species diversity, which also considered the effects of species richness and community evenness. The Shannon index of the hibernation group was significantly higher than that of the control group (*p* < 0.05), indicating that the richness and evenness of the gut microbiota in the hibernation group were improved (Figure 4C). We also found coverage above 0.99 for all samples from wild chipmunks in both groups, indicating that the sequencing depth was sufficient to cover most microorganisms, including rare species.

### 3.3. Beta Diversity Analysis

Beta diversity analysis was used to compare differences and similarities in species diversity between the hibernation group and the control group. In this study, principal coordinate analysis (PCoA) using the feature-based weighted Bray–Curtis algorithm showed obvious clustering separation (Figure 5A). Further PERMANOVA analysis showed (Figure 5B) that there was a significant difference in beta diversity of the gut microbiota between the hibernation and control groups and that experimental grouping was meaningful (R^2^ = 0.501, *p* = 0.001). A heatmap was drawn up based on the unweighted UniFrac algorithm to obtain the distance matrix between samples, using R language. As shown in Figure 5C, the composition of the three samples in the hibernation group was highly similar and closer, which was different from the three samples in the control group.

### 3.4. Composition Analysis of Gut Microbiota

The gut microbiota of wild chipmunks included 2 kingdoms, 33 phyla, 75 classes, 183 orders, 316 families, 523 genera, and 610 species. According to the annotation results, the top 10 species in each level were selected and the rest were combined as others to draw the bar chart of species distribution. Unclassified represents the species that have not been taxonomically annotated.

At the phylum level (Figure 6A), the top 10 phylas in relative abundance were Bacteroidetes, Firmicutes, Proteobacteria, Campylobacterota, Acidobacteriota, Deferribacterota, Actinobacteriota, Verrucomicrobiota, Gemmatimonadota, and Chloroflexi. The dominant bacteria in the hibernation group were Firmicutes (21.44%), Bacteroidetes (21.16%), Proteobacteria (11.30%), and Campylobacter (9.74%). The dominant bacteria in the control group were Bacteroidetes (56.31%), Firmicutes (32.53%), Verrucomicrobia (5.00%), and Proteobacteria (3.29%).

As shown in Figure 6E, the dominant genera of the hibernation group included Helicobacter (9.73%), unclassified Muribaculaceae (8.59%), and Mucispirillum (6.80%). The dominant genera in the control group included Prevotellaceae (14.45%), unclassified Muribaculaceae (9.70%), Erysipelatoclostridium (8.47%), unclassified Prevotellaceae (7.15%) and Prevotella-9 (6.20%), and the average proportion of other genera was lower (<5%).

### 3.5. Structural Differences in Gut Microbiota Analysis

By comparing the relative abundance of the gut microbiota of wild chipmunks in different periods, the analysis was performed using the Wilcoxon rank sum test, and the test results are shown in Figure 7. At the phylum level (Figure 7A), compared with the control group, the hibernation group had a significant reduction in the relative abundance of Bacteroidetes and Firmicutes (*p* < 0.05) and a significant increase in the relative abundance of Proteobacteria, Campylobacter, Acidobacteria, Deferriobacterota, Actinobacteriota, and Gemmatimonadota (*p* < 0.05). At the genus level (Figure 7B), the dominant genera Helicobacter and Mucispirillum were significantly increased in the hibernation group (*p* < 0.05). The relative abundance of prevotellaceae-UCG-001, Erysipelatoclostridium, unclassified Prevotellaceae, and Prevotella-9 was significantly decreased (*p* < 0.05). To further explore the structural differences of the gut microbiota in wild chipmunks during hibernation, the linear discriminant analysis effect size (LEfSe) method was used in this study. As shown in Figure 8 and Figure 9, at the species level, the relative abundance of Helicobacter typhlonius, Mucispirillum schaedleri, unclassified Allobaculum, and other species was significantly increased in the hibernation group, while a significant decrease in relative abundance was observed for species such as unclassified Prevotella-9, unclassified Prevotellaceae-UCG-001, unclassified Prevotellaceae-UCG-003, and other species of Prevotella. The marker intergroup abundance histogram shows the differences between the dominant microbiota in the two groups of chipmunks at the species level more intuitively (Figure 9).

### 3.6. PICRUSt2 Functional Prediction Analysis

Based on the high-throughput sequences, PICRUSt2 was used in the analysis of function prediction in the second level KEGG pathway of the wild chipmunks. As shown in Figure 10, there were 27 functional pathways with significant differences between the hibernation and control groups (*p* < 0.05). PICRUSt2 functional prediction analysis of the gut microbiota showed that 15 pathways, such as lipid metabolism, xenobiotics biodegradation and metabolism, amino acid metabolism, environmental adaptation, and neurodegenerative diseases, were significantly enriched in the hibernation group, while 12 pathways, including carbohydrate metabolism, replication and repair, translation, nucleotide metabolism, transcription, and biosynthesis of other secondary metabolites, were significantly enriched in the control group.

## 4. Discussion

The hibernation pattern of wild chipmunks is different from that of traditional hibernators, which store only food or fat before hibernation [36]. Their unique mixed hibernation strategy may lead to different changes in colonic epithelial tissue and the gut microbiota during hibernation. This study conducted histological analysis of the colonic epithelial tissue, compared the diversity and composition of the gut microbiota in wild chipmunks during hibernation and non-hibernation stages, and predicted the functional changes of the gut microbiota using 16S rRNA gene high-throughput sequencing technology.

Histological analysis of the colonic epithelial tissue showed that hibernation caused the increase of goblet cells in the colonic epithelial tissue of wild chipmunks. Previous studies have shown that long-term fasting led to an increase in the number of duodenal goblet cells in rats [48]. Additionally, both carp [49] and chickens [50] showed the increase of goblet cells during fasting. Goblet cells secrete mucin, which is involved in the formation of the mucous layer on the intestinal epithelium, protecting it from pathogens and playing an essential role in maintaining intestinal barrier function [51,52]. The mucin secreted by goblet cells is considered the first line of defense against intestinal inflammation, and its decrease is a major risk factor for intestinal inflammation [53,54]. Hibernation fasting can lead to increased intestinal permeability and disruption of intestinal barrier function [55], while the increase of goblet cells can promote mucin production, which may serve as a protective compensatory response to reduce the entry of toxic substances and inflammatory metabolites into the body circulation during hibernation. Prolonged fasting during hibernation inhibits epithelial cell proliferation, causing extensive villi shedding in the intestinal epithelium [56], and goblet cells can enhance the stability and tolerance of the intestinal tract [57]. Goblet cell hyperplasia might compensate for the inhibitory effects of epithelial cell proliferation and enhance the stability of the intestinal environment of wild chipmunks during hibernation. Interestingly, in another study of Daurian ground squirrels during hibernation, it was found that hibernation reduced the number of goblet cells and disrupted some microvillous structures in the colonic tissue of Daurian ground squirrels [21]. The difference in the goblet cells between the wild chipmunks and the Daurian ground squirrels may be related to species differences.

Alpha diversity analysis showed that ACE, Chao1, and Shannon indices were significantly higher in the hibernation group than in the control group, and that wild chipmunks showed a significant increase in the richness and diversity of their gut microbiota during hibernation. This is in agreement with previous findings on Siberian chipmunks [36], dwarf lemurs [25], and bumblebee queens [58] during hibernation. The reason may be that the increased abundance of the gut microbiota during hibernation contributes to the metabolic processes and energy reserves of the wild chipmunk. However, hibernation has reduced the richness and diversity of gut microbiota in studies of 13-lined ground squirrels [18], brown bears [26], and *Strauchbufo raddei* [29], and the reason may be related to species differences and dietary differences. Beta diversity analysis based on the Bray–Curtis distance matrix showed a clear clustering separation, with samples from the hibernation and control groups each clustering well into one group, and there were significant differences in the gut microbiota diversity of the two groups of chipmunks. 

Structural differences in gut microbiota analysis showed that in agreement with those of other rodents, such as 13-lined ground squirrels [59], arctic ground squirrels [22], and BALB/c mice [60], the gut microbiota of the wild chipmunk is mainly composed of Bacteroidetes, Firmicutes, and Proteobacteria. This is equally consistent with amphibians such as Japanese wrinkled frogs [28] *Strauchbufo raddei* [29] and *Fejervarya limnocharis* [30], and brown tree frogs [16] as well as other mammals such as snub-nosed monkeys [61] and brown bears [26]. In the hibernation group, the relative abundance of both Bacteroidetes and Firmicutes was significantly decreased, while the relative abundance of Proteobacteria was significantly increased. Bacteroidetes belongs to the Gram-negative bacterial group and is mainly associated with degradation of polysaccharides and carbohydrates [62]. Firmicutes belongs to the Gram-positive bacterial group; it can use carbohydrates, polysaccharides, and fatty acids as energy sources [63], and it can also ferment carbohydrates to produce short-chain fatty acids, which promote the health of the host [64]. The reduced relative abundance of both Bacteroidetes and Firmicutes may be related to the shift of energy metabolism from carbohydrate metabolism to lipid metabolism of the wild chipmunks in hibernation. Proteobacteria was found to have the potential to contribute to the development of obesity, and isolation of Proteobacteria from an obese population led to the development of obesity in germ-free mice [65]. The elevated relative abundance of Proteobacteria during the hibernation of wild chipmunks corresponds to the increased fat reserves of the wild chipmunks in response to the hibernation challenge. The ratio of Firmicutes/Bacteroidetes (F/B) was elevated in the hibernation group compared to the control group. A previous study found that mice were observed to have elevated F/B under the influence of intermittent fasting [66], which is similar to the hibernation pattern of chipmunks, which also fast intermittently. Elevated F/B was found to contribute to the promotion of fat accumulation [67,68]. The composition of gut microbes is an important factor influencing energy storage, and an obesity-associated gut microbiota is more conducive to the host’s ability to derive energy from the diet [69]. We speculate that the altered levels of gut microbiota phylum observed during hibernation may function in allowing the host to better obtain nutrients from the intestines, to rapidly acquire and utilize energy, to store fat, and to better adapt to the severe trials associated with hibernation. At the genus level, the relative abundance of the dominant bacteria *Helicobacter* and *Mucispirillum* was significantly higher in the hibernation group. By contrast, there was a significant decrease in the relative abundance of *prevotellaceae-UCG-001*, *Erysipelatoclostridium*, *unclassified Prevotellaceae*, and *Prevotella-9*. The relative abundance of *Helicobacter typhlonius*, a species of *Helicobacter*, was significantly increased in the hibernation group. *Helicobacter typhlonius* was first identified in the colon of IL-10-deficient mice. It was urease-negative [70], and belonged to the non-gastric Helicobacter pylori group, growing in the intestine rather than in the stomach. It can elicit a strong T-cell response, promote immune cell proliferation differentiation [71], and induce intestinal inflammation [72]. The increase in *Helicobacter typhlonius* suggests that the gut microbiota may contribute to intestinal homeostasis during chipmunks’ hibernation by modulating the intestinal immune system. *Mucispirillum schaedleri*, a specialized anaerobic bacterium that colonizes the intestinal mucus layer [73], has been shown to have the ability to modulate host immunity as well as interfere with the invasion of pathogens [74]. In the hibernation group, the significant increase in the relative abundance of *Mucispirillum schaedleri* suggests that the intestinal immune function of wild chipmunks may be enhanced during hibernation by increasing the level of the relative abundance of *Mucispirillum schaedleri*, which maintains the stability of the intestinal internal environment of chipmunks during hibernation. *Prevotella* is closely associated with the catabolism of carbohydrates as well as plant polysaccharides [75]. *Prevotella* has a pro-inflammatory effect, and the increase in intestinal *Prevotella* leads to an increased incidences of arthritis and colitis [76]. It also plays a role in the degradation of mucus proteins [77], which may lead to an increase in intestinal permeability. *Unclassified Prevotella-9*, *unclassified Prevotellaceae*, *unclassified Prevotellaceae-Ga6A1-group*, *unclassified Prevotellaceae-UCG-001*, and *unclassified Prevotellaceae-UCG-003*, all belong to *Prevotella*. The decrease in the relative abundance of *Prevotella* during hibernation in the colonic microbiota of chipmunks might reduce the incidence of intestinal inflammation and better maintain the homeostasis of the intestinal internal environment as well as the health status of the organism during hibernation.

PICRUSt2 functional prediction analysis showed that the lipid metabolism pathway was significantly enriched in the hibernation group and that the carbohydrate metabolism pathway was significantly enriched in the control group, suggesting that the colonic microbiota of wild chipmunks undergoes a shift during hibernation from functional taxa related to carbohydrate metabolism to functional taxa related to lipid metabolism. This may be associated with the utilization and conversion of substrates involved in energy metabolism during hibernation. Consistent with previous studies, a study on the great horseshoe bat (GHB) demonstrated that the metabolic function of the gut microbes of the GHB switched from a carbohydrate-related function to a lipid-related functional category during hibernation [78]. Similarly, in a study of the Japanese black bear’s liver, genes related to glycolysis and amino acid catabolism were found to be downregulated during hibernation, while genes involved in lipid metabolism and gluconeogenesis were upregulated [79]. All these indicate that the metabolic mode of animals during hibernation shifts from carbohydrate metabolism to lipid metabolism, and lipids are the main energy source during hibernation. The exogenous substance metabolism and degradation pathway were significantly enriched in the hibernation group, which is compatible with the unique hibernation pattern of wild chipmunks: intermittent feeding and drinking during hibernation [35]. Amino acid metabolism pathways were also significantly enriched in the hibernation group, which resulted in a consequent increase in the number of substrates related to amino acid metabolism in the gut, which in turn corresponded to the structural changes in the gut microbiota involved in amino acid metabolism. The significant enrichment of amino acid metabolic pathways in the hibernation group has been suggested to be possibly related to muscle wasting during hibernation of chipmunks [80]. In most rodents, muscle mass and protein content decrease during hibernation [81], but this is not as severe as in traditional muscle models, since muscle breakdown is limited. A study on hibernation of mammals showed that muscle mass and protein values decreased by an average of 15.3% and 7.7% after hibernation compared to pre-hibernation [82]. The significant enrichment of the neurodegenerative diseases pathway in the hibernation group suggests an adaptive change in the inhibition of neuroexcitation in wild chipmunks during hibernation. Low temperature has been found to cause damage to synaptic structures in hibernating mammals, but this alteration is reversible upon rewarming [83], and the enrichment of the neurodegenerative diseases pathway in the hibernation group provides a clue for the treatment of neurodegenerative diseases by cooling [84,85,86,87]. Enrichment of the environmental adaptation pathway in the hibernation group suggests that wild chipmunks undergo a shift in their gut microbiota towards taxa that are more adaptive to environmental changes in order to cope with the extreme environment imposed by hibernation. The pathways enriched in the control group were mainly involved in a range of basic life functions, including replication and repair, transcription, translation, biosynthesis of other secondary metabolites, and nucleotide metabolism. They were significantly reduced in the hibernation group, indicating that the gut microbiota is involved in the regulation of basal vital activities of the organism, and that hibernation causes a reduction in the level of basal vital activities of wild chipmunks. It has been shown that the levels of basal metabolic indicators such as metabolic rate and oxygen consumption decrease during hibernation in mammals [88,89].

## 5. Conclusions

In summary, hibernation caused the increase of goblet cells in the colonic epithelial tissue of wild chipmunks, and increased the richness and diversity of the gut microbiota. The relative abundance of Bacteroidetes and Firmicutes decreased significantly, while the relative abundance of Proteobacteria increased significantly, with an increased ratio of Firmicutes/Bacteroidetes (F/B). The relative abundance of *helicobacter typhlonius* and *Mucispirillum schaedleri* in the colon of wild chipmunks increased significantly during hibernation. The relative abundance of *unclassified Prevotella-9*, *unclassified Prevotellaceae-UCG-001*, *
unclassified Prevotellaceae-UCG-003*, and other species of *Prevotella* decreased significantly. The pathways of the gut microbiota in the hibernation group were mainly enriched in lipid metabolism, xenobiotics biodegradation, metabolism amino acid metabolism, environmental adaptation, and neurodegenerative diseases, as shown by PICRUSt2 analysis. In conclusion, the diversity and richness of the gut microbiota were increased and the composition of the gut microbiota was changed during hibernation. The changes of the histology of colonic epithelial tissue, and composition and function of gut microbiota could regulate the physiological metabolism of wild chipmunks during hibernation and better maintain their normal physiological functions.

## Figures and Tables

**Figure 1 animals-14-01498-f001:**
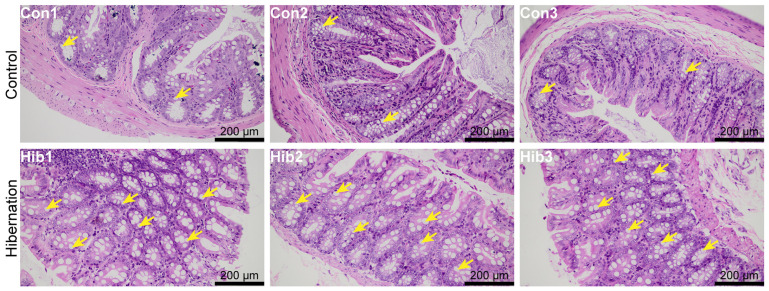
Hematoxylin-eosin (HE) staining of colonic epithelial tissue of wild chipmunks. (Con 1, 2, 3) Control group; (Hib 1, 2, 3) hibernation group. Goblet cells (GCs) are indicated with yellow arrows.

**Figure 2 animals-14-01498-f002:**
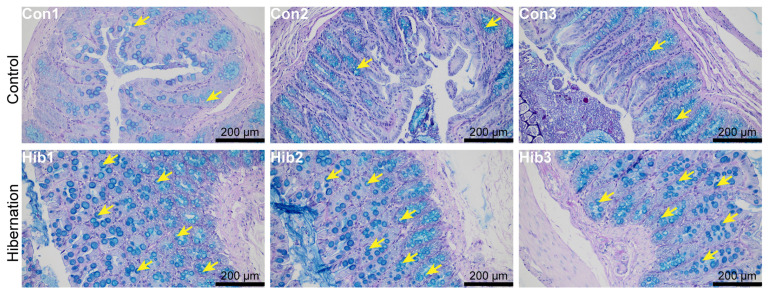
Alcian blue–periodic acid–Schiff (AB-PAS) staining of colonic epithelial tissue of wild chipmunks. (Con 1, 2, 3) Control group; (Hib 1, 2, 3) hibernation group. Goblet cells (GCs) are indicated with yellow arrows.

**Figure 3 animals-14-01498-f003:**
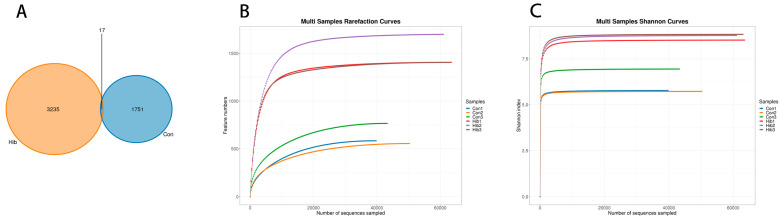
Venn diagram. (**A**) The overlapping part represents the number of common features, and the non-overlapping part represents the number of unique features. (Con) Control group; (Hib) hibernation group. (**B**) Rarefaction curves. *X*-axis: Counts of randomly sampled sequences; *Y*-axis: Counts of features detected by given sequences. (**C**) Shannon index curves. *X*-axis: Counts of randomly sampled sequences; *Y*-axis: corresponding Shannon index.

**Figure 4 animals-14-01498-f004:**
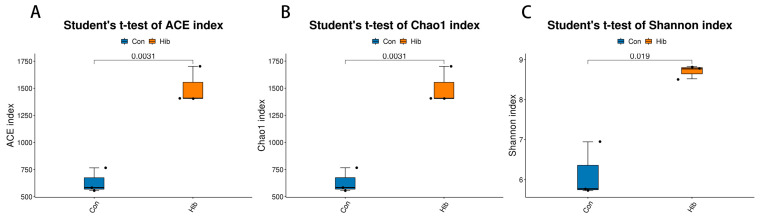
Boxplot of alpha diversity indices. (**A**) ACE index. (**B**) Chao 1 index. (**C**) Shannon index. *X*-axis: Group name; *Y*-axis: Alpha diversity indices. The number of lines between bars is the *t*-test *p* value. (Con) Control group; (Hib) hibernation group.

**Figure 5 animals-14-01498-f005:**
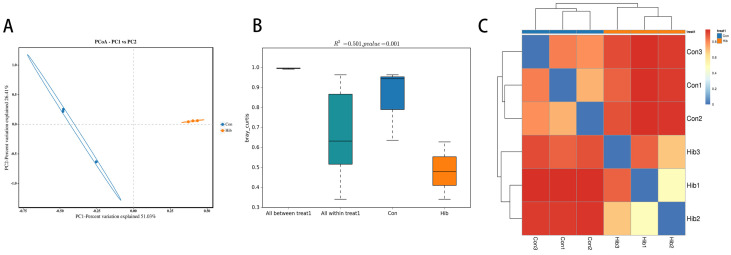
Beta diversity analysis of the wild chipmunks. (**A**) Principal coordinate analysis (PCoA) graph. PCoA1 explained 51.03% and PCoA2 explained 26.41% of the total variation of the samples. (Con) Control group; (Hib) hibernation group. (**B**) PERMANOVA analysis box plot. The box above ‘All between’ represents the beta distance data of samples between all groups, while the box above ‘All within’ represents the beta distance data of samples within all groups. (**C**) Sample distance clustering heatmap. The color gradient from blue to red indicates the distance between samples from close to far.

**Figure 6 animals-14-01498-f006:**
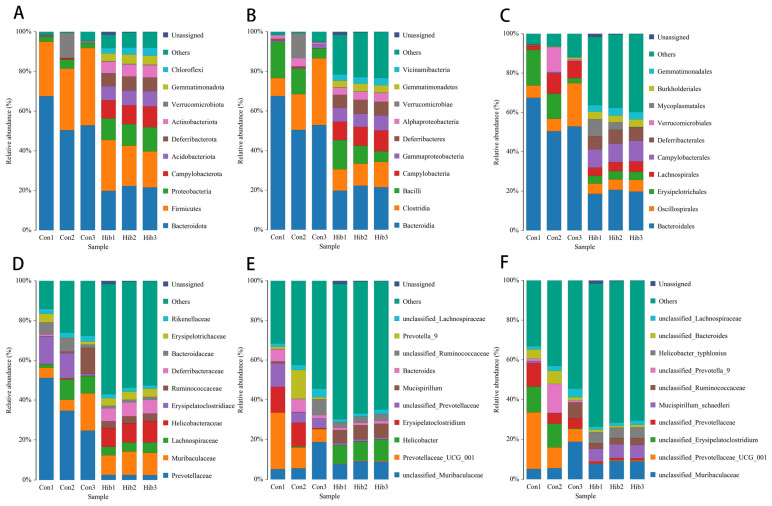
Composition of the gut microbiota in the wild chipmunk. (**A**) Phylum. (**B**) Class. (**C**) Order. (**D**) Family. (**E**) Genus. (**F**) Species. (Con 1, 2, 3) Control group; (Hib 1, 2, 3) hibernation group.

**Figure 7 animals-14-01498-f007:**
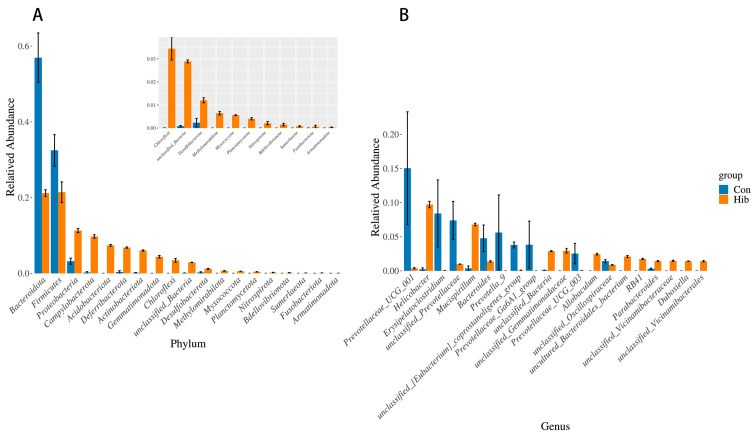
Wilcoxon rank sum test analysis histogram. (**A**) Phylum level; (**B**) genus level. *X*-axis represents species (the top 20 species with the lowest p values are shown, *p* < 0.05); *Y*-axis represents the relative richness of species. (Con) Control group; (Hib) hibernation group.

**Figure 8 animals-14-01498-f008:**
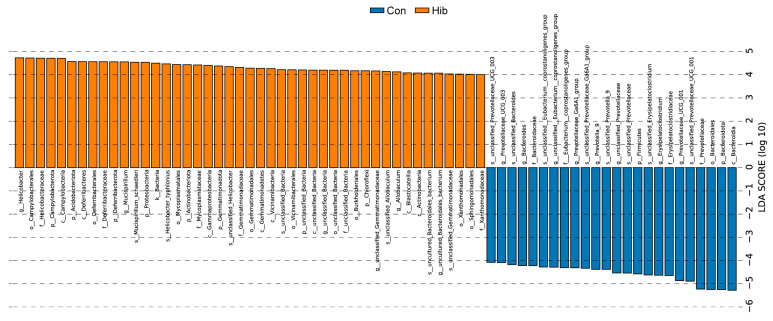
LDA score distribution histogram. Y-axis: Features that show significant difference between groups, Phylum (p), Class (c), Order (o), Family (f), Genus (g), and Species (s); *X*-axis: Log10 of LDA score, where LDA > 4 is considered to be significantly different. (Con) Control group; (Hib) hibernation group.

**Figure 9 animals-14-01498-f009:**
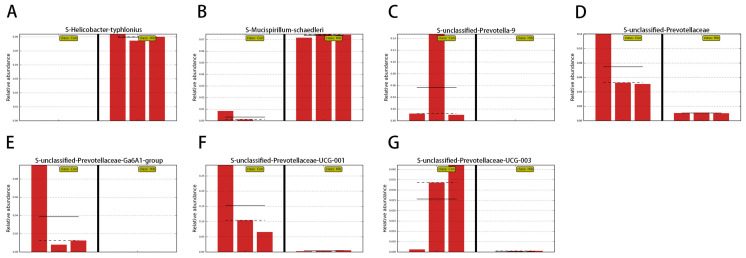
Marker intergroup abundance histogram. The solid and dash lines represent the average and median of relative abundance. (**A**) Helicobacter typhlonius. (**B**) Mucispirillum schaedleri. (**C**) Unclassified Prevotella-9. (**D**) Unclassified Prevotellaceae. (**E**) Unclassified Prevotellaceae-Ga6A1-group. (**F**) Unclassified Prevotellaceae-UCG-001. (**G**) Unclassified Prevotellaceae-UCG-003. (Con) Control group; (Hib) hibernation group.

**Figure 10 animals-14-01498-f010:**
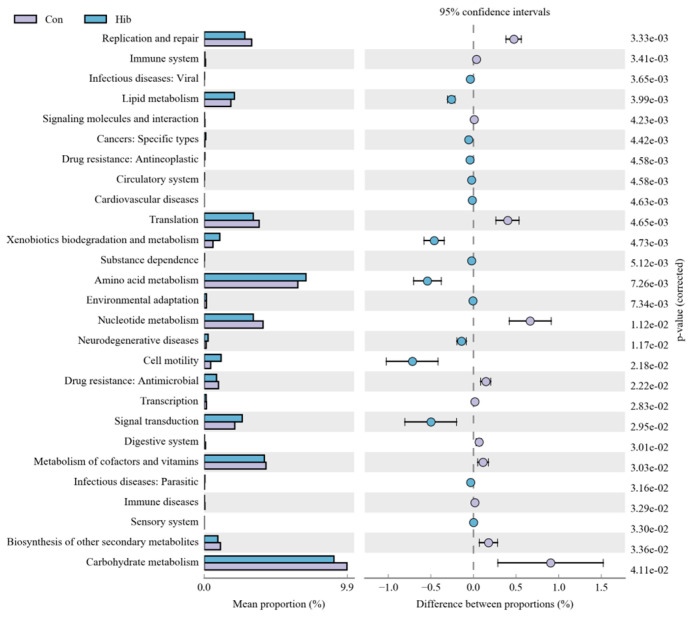
Differential analysis of function prediction in the second level KEGG pathway of the wild chipmunks. (Con) Control group; (Hib) hibernation group.

## Data Availability

The study’s original contributions are included in the article. Further inquiries can be directed to the corresponding authors.

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
