# Peer review of "Effects of Hibernation on Colonic Epithelial Tissue and Gut Microbiota in Wild Chipmunks (Tamias sibiricus)"

_animals, 2024, doi:10.3390/ani14101498_

Round 1
Reviewer 1 Report
Comments and Suggestions for Authors
This is a novel and meticulous article on the effect of the cecal tissue and gut microbiota of wild chipmunks amid adaptive changes induced by hibernation. The topic is interesting. There are a few things that are worth noting.
1、The abstract needs to be shortened.
2、Only three samples for each group, it is really small for analyzing gut microbiota. Thus, the results are not very convincing.
3、There are many studies on the effects of hibernation on gut microbiota in the Daurian ground squirrel, which belongs to a type of ground squirrel. Therefore, I suggest that it can be described in the Introduction.
4、One of the innovations mentioned in lines 99-104 is that the research object was the wild chipmunk in northeast China. There have been some studies on the effects of hibernation on gut microbiota in Siberian chipmunks. Then, are there any similarities or differences between the Siberian chipmunk and those from northeast China? I think this is what researchers want to see, and I suggest explaining it in the article.
5、Another innovative point of the article is using histological analysis to show that the goblet cells of wild chipmunks had a significant increase during hibernation, can we add 1-2 additional H&E plots of other samples to enhance accuracy and credibility?
6、In addition, it is recommended to indicate the scale bar of the hibernation group as well in Figure 1.
7、Fig.2, Fig.4, Fig.5, and Fig.7 are of low resolution and hard to read, higher resolution is needed.
8、Please carefully check the citations for accuracy and completeness to ensure that relevant literature is cited wherever necessary.
Comments on the Quality of English LanguageModerate editing of the English language required.
Author Response
Thank you for taking the time to review our manuscript. We appreciate your valuable feedback and constructive suggestions. Below, we have addressed each of your comments and provided explanations or revisions accordingly.

Reviewer 2 Report
Comments and Suggestions for Authors
In the current article, the authors undertook a 16S rRNA gene amplicon sequencing method to analyse the gut microbiome of hibernating wild chipmunks. They further correlated their results with histological analysis of the large intestine (cecum) of the animals studied. The paper sheds new light on the role of bacteria in intestinal regulatory processes in an original way. Overall, it is reasonably well conceived and methodologically correctly executed.
Critical comments:
1. it should be borne in mind that from a morphological (histological) point of view (and this is, after all, the aspect undertaken by the authors in their paper) there are 4 tissues: epithelial, connective, muscular and nervous. Therefore, the use of the terms ‘cecal tissue’, ‘gut tissue’ or others is incorrect. Please amend this.
2. line 119 - please describe the euthanasia protocol (doses, veterinary supervision, etc.) in detail.
3. since the animals were in laboratory conditions for more than 120 days (14 days of adaptation and 120 days of hibernation) it is hard to consider them ‘wild’. How do the authors prove that the adaptation period (food and water intake) did not affect the composition of the gut microbiome?
4 A detailed justification is required as to why only one section of the large intestine was taken? What about the colon and rectum?
5. please provide information on primer design and validation.
6. Figure 1 is not informative in any way. Firstly, it is unclear what the authors wanted to convey through this figure? Secondly, with each figure the description should be self-explanatory and leave no doubt of interpretation. Thirdly, I do not see the point of inserting pairs of photographs (original and enlargement). The enlargement alone is sufficient. Fourthly, the arrows are only visible in the Hib group which suggests the absence of goblet cells in the Con group!
Author Response

(The authors gave the same response as above.)

Round 2
Reviewer 1 Report
Comments and Suggestions for Authors
This manuscript could be accepted.
Comments on the Quality of English LanguageNone